# Semi-Classical Einstein Equations: Descend to the Ground State

**Zbigniew Haba** 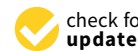

Institute of Theoretical Physics, University of Wroclaw, Plac Maxa Borna 9, 50-204 Wroclaw, Poland;
zbigniew.haba@uwr.edu.pl

**Abstract:** The time-dependent cosmological term arises from the energy-momentum tensor calculated in a state different from the ground state. We discuss the expectation value of the energy-momentum tensor on the right hand side of Einstein equations in various (approximate) quantum pure as well as mixed states. We apply the classical slow-roll field evolution as well as the Starobinsky and warm inflation stochastic equations in order to calculate the expectation value. We show that, in the state concentrated at the local maximum of the double-well potential, the expectation value is decreasing exponentially. We confirm the descent of the expectation value in the stochastic inflation model. We calculate the cosmological constant $\Lambda$ at large time as the expectation value of the energy density with respect to the stationary probability distribution. We show that $\Lambda \simeq \gamma^{\frac{4}{3}}$ where $\gamma$ is the thermal dissipation rate.

**Keywords:** inflation; false vacuum; cosmological constant

## 1. Introduction

In the description of inflation, it is understood that at the early stages of the universe evolution quantum physics is relevant. It is not clear how to take quantum effects into account because the Einstein equations are classical. For later evolution leading to the structure formation, the quantization of gravity and the scalar field in a linear approximation around the homogeneous classical solution leads to satisfactory results [1–4]. The standard approach [4,5] to inflation driven by the scalar field is in fact semi-classical. It is assumed that the classical scalar field evolution arises in a quantum field theory in the classical limit [6]. The quantum fluctuations as a result of inflation lose their quantum character and behave as classical fluctuations [7,8]. The quantum statistical mechanics is applied in order to use the result that the quantum system evolves to its ground state. For a typical double-well potential, the quadratic approximation has a negative mass square term. Such a quadratic Lagrangian has a local maximum at $\phi = 0$. The classical system starting close to $\phi = 0$ will quickly evolve to $\phi \neq 0$. The same happens in quantum mechanics [6]. The Gaussian state localized at $\phi = 0$ evolves exponentially fast to a localization at $\phi \neq 0$. In another description [9–11], one describes the state evolution from one side of the barrier $\phi < 0$ to another side with $\phi > 0$ as the barrier penetration. Then, one can calculate the resulting wave function [11] by semi-classical imaginary time functional integral. By Einstein equations, the universe evolution is driven by the expectation value of the energy-momentum. The different speed of the expansion of the universe can be explained by the time dependence of this expectation value (time dependent cosmological term). In other words, the time dependence of the cosmological term can be associated with the evolution from the false vacuum to the ground state [12–14]. From CMB observations, we know that the universe was once (before reionization) in the thermal state. It may be that such a thermal state appeared already at an earlier stage of universe evolution [15,16]. The barrier penetration takes place also in thermal systems [17]. For a quantum description, we need the quantum mechanics in a thermal state.

In this paper, we consider a method of the description of the vacuum decay based on the slow-roll and diffusion approximations to the quantum evolution in an expanding universe [18,19]. This is a long wave approximation which neglects the second-order derivatives in equations of motion and first-order derivatives in the energy-momentum tensor. The noise in the stochastic equation consists of two independent parts: the quantum noise of Starobinsky [18] and Vilenkin [20] and the thermal noise describing thermal fluctuations [19,21–23]. We consider Einstein equations with the expectation value of the energy-momentum tensor in a quantum state $\psi$ on the rhs of these equations. There is an unspecified problem of the choice of $\psi$ and a difficult task to determine its time evolution on a non-perturbative level (to treat a transition between states which are not related by perturbation theory). We wish to investigate the time evolution of the expectation value of the energy density in a state simulating the false vacuum and for a comparison in a state close to the true ground state. The aim is to study the behavior of the cosmological term during the transition from the false vacuum to the ground state. We choose as a test state first the state localized near the local maximum of the potential and subsequently a state localized close to the minimum of the potential. The time evolution of the states is equivalent to the time evolution of the fields defining the energy-momentum. First, we obtain this time evolution in a classical and slow-roll approximation. Then, the expectation value is calculated. It is shown that the expectation value in the false vacuum is decreasing exponentially in time. In the next step, we are interested in the expectation value in the non-perturbative thermal quantum state. According to the authors of [18,19,22,23], in a slow roll approximation, the field evolution in such a quantum state can be obtained as a solution of a stochastic equation with quantum and thermal noise. We discuss approximate solutions of such equations. We conclude that the approximate solutions have the same qualitative (exponential) behavior as the classical solutions. We are unable at the moment to treat precisely the non-linear stochastic equations or the Fokker–Planck partial differential equations for the probability distribution of the stochastic process. The large time behavior is determined by the equation for stationary probability distribution. We can obtain an exact solution of this equation. This allows calculating the expectation values at large time. The classical slow-roll approximation is not reliable for large time. We showed in [24] that the stochastic version (because of quantum and thermal fluctuations) can make sense for arbitrarily large time. Then, the evaluation of the mean value of the energy density may give some qualitative explanation why the cosmological term is big at the early time of inflation and it is decreasing to a small value at large time. The same model when applied in the $\delta N$ formalism (initiated in [3,25] and developed in [26]) can be used for a calculation of the spectrum of quantum and thermal fluctuations.

The paper is organized as follows. In Section 2, we define the semi-classical approximation for Einstein equations. In Section 3, we discuss the slow-roll stochastic approximation for quantum evolution in a thermal state. In Section 4, we calculate the energy-density in the absence of noise. In Section 5, the approximate stochastic equations are solved. The asymptotic behavior of expectation values is discussed in Section 6 where the stationary probability for the stochastic process is calculated. In particular, we show that the cosmological constant at large time is proportional to the thermal diffusion constant.

## 2. The Semi-Classical Einstein Equations

First, we quantize the scalar field in an external metric. Subsequently, we determine the metric from the Einstein equations

$$R_{\mu\nu} - \frac{1}{2}g_{\mu\nu}R = 8\pi G < \psi|T_{\mu\nu}(\phi_t)|\psi > . \tag{1}$$

Here, $G$ is the Newton constant, $\phi_t$ is the quantum scalar field in an external metric $g_{\mu\nu}$, and $|\psi >$ is the quantum state of the field $\phi$. It is understood that Equation (1) comes from an average of a complete inflationary model like that of Ref. [4]. The quantization of renormalizable interactions in an external metric is well-defined on the perturbative level [27]. The expectation values of the

energy-momentum are usually calculated in the Bunch–Davis vacuum of the free field theory [27]. They require renormalization because of the ultraviolet divergences. The expectation value and renormalization add new terms to the lhs of the Einstein equations (Equation (1)) [28,29]. In this paper, we separate the long wave and short wave problems, restricting ourselves to the study of the latter. To discuss the ground state of an interacting field theory, we need to go beyond the perturbation theory.

We would also be interested in an expectation value of the energy-momentum tensor $T_{\mu\nu}$ on the rhs of Equation (1) in a mixed state $\rho$, i.e.,

$$R_{\mu\nu} - \frac{1}{2}g_{\mu\nu}R = 8\pi G Tr\left(\rho T_{\mu\nu}(\phi_t)\right). \tag{2}$$

In the slow-roll classical approximation, the expectation value of Equation (1) is

$$< \psi|T_{\mu\nu}(\phi_t(\phi))|\psi > = \int d\phi |\psi(\phi)|^2 T_{\mu\nu}(\phi_t(\phi)), \tag{3}$$

where $\phi_t(\phi)$ is the solution of the equation

$$3H\partial_t\phi = -V'(\phi). \tag{4}$$

with the initial condition $\phi$. It follows from the Einstein equations (Equation (1), in the Friedman form) that in a flat homogeneous expanding universe

$$H^2 = \frac{8\pi G}{3}V(\phi) \tag{5}$$

in the slow-roll approximation.

## 3. Stochastic Equations for Slow-Roll Inflation

Observations of the CMB radiation show that the universe is spatially flat. We consider a flat (homogeneous) expanding metric

$$ds^2 = g_{\mu\nu}dx^\mu dx^\nu = dt^2 - a(t)^2 d\mathbf{x}^2.$$

In this metric, the quantum field behaves as a classical diffusion process [18,20] with the noise $\frac{3}{2\pi}H^{\frac{5}{2}}\partial_t W$ [30] where $W$ is the Brownian motion [31]. We consider a stochastic wave equation with friction

$$\partial_t^2\phi - a^{-2}\triangle\phi + (3H + \beta\gamma^2)\partial_t\phi + V'(\phi) + \frac{3}{2}\beta\gamma^2 H\phi = \eta, \tag{6}$$

where $\beta^{-1}$ is the temperature of the environment, $H = a^{-1}\partial_t a$ and $\eta$ is a noise. Further on, we make the slow-roll approximation neglecting $\partial_t^2\phi$ term. We assume that the friction $\beta\gamma^2$ is small in comparison to $3H$. We also neglect the spatial derivatives ($\triangle\phi$). We can take into account thermal effects by an addition of the thermal noise to the rhs of Equation (1). Now,

$$\eta = \gamma a^{-\frac{3}{2}}\partial_t B + \frac{3}{2\pi}H^{\frac{5}{2}}\partial_t W, \tag{7}$$

where the first term describes the thermal noise and the second term the quantum noise. The thermal part of this equation is derived in [32,33]. The factor $a^{-\frac{3}{2}}$ comes from $\det|g_{\mu\nu}|^{\frac{1}{4}}$ and the factor $H^{\frac{5}{2}}$ in the Starobinsky noise is chosen to reproduce the correlation functions of the quantum scalar field in an expanding universe [30]. The (white) noise $\partial_t B$ is the Gaussian random process with the covariance

$$\langle\partial_t B\partial_s B\rangle = \delta(t - s). \tag{8}$$

$\partial_t W$ is an independent Gaussian stochastic process with the same covariance (Equation (8)). Equation (1) with quantum and thermal noise is discussed in [21–23].

We interpret Equation (6) as a stochastic equation in the Stratonovitch sense [31] (we apply the stochastic calculus and the ∘-circle notation of [31] for stochastic multiplication). In the slow-roll approximation and small friction ($\gamma^2 \simeq 0$),

$$3Hd\phi = -V'dt + \gamma a^{-\frac{3}{2}} \circ dB + \frac{3}{2\pi}H^{\frac{5}{2}} \circ dW. \tag{9}$$

If $H$ is known (from Equation (5)), then in principle we can determine $a(\phi)$ after solving the stochastic Equations (1) and (6). We can obtain an explicit formula if we neglect the noise and apply the slow roll approximation in Equation (9) (without noise). Then,

$$\ln(a) = \int Hdt = \int d\phi (\frac{d\phi}{dt})^{-1}H = -8\pi G \int d\phi V(V')^{-1}. \tag{10}$$

We can generalize the stochastic Equation (9) to multiple scalar fields [25,34] $\phi = (\phi^1, ..., \phi^D)$ as in the standard model of weak and strong interactions with the Higgs potential

$$V = \frac{g}{4}(|\phi|^2 - \frac{\mu^2}{g})^2.$$

Then, the noise $\eta = (\eta^1, ...., \eta^D)$ consists of independent random Gaussian variables with the same variance. In Equation (9) $\phi$ and the noises should be treated as vectors, $V' \to \nabla V$ and $(\partial_t \phi)^2 \to |\partial_t \phi|^2$ (the length of the vector $\partial_t \phi$). We can calculate $a(\phi)$ if $V = V(|\phi|)$ is rotation invariant with $|\phi|^2 = (\phi^1)^2 + ... + (\phi^D)^2$. Then (after an omission of noise in Equation (9)),

$$\ln(a) = \int Hdt = \int d|\phi|(\frac{d|\phi|}{dt})^{-1}H = -8\pi G \int d|\phi|V\left(\frac{dV}{d|\phi|}\right)^{-1}. \tag{11}$$

We have the Fokker–Planck equation [35] for the probability distribution of $\phi$ (with the Stratonovich interpretation of the stochastic equations [31])

$$\partial_t P = \frac{\gamma^2}{18}\partial_\phi \frac{1}{Ha^{\frac{3}{2}}}\partial_\phi \frac{1}{Ha^{\frac{3}{2}}}P + \frac{1}{8\pi^2}\partial_\phi H^{\frac{3}{2}}\partial_\phi H^{\frac{3}{2}}P + \partial_\phi(3H)^{-1}V'P. \tag{12}$$

In the multifield case ($\phi \in R^D$) with $\partial_j = \frac{\partial}{\partial\phi^j}$, Equation (12) reads

$$\partial_t P = \sum_j \partial_j \left(\frac{\gamma^2}{18}\frac{1}{Ha^{\frac{3}{2}}}\partial_j\frac{1}{Ha^{\frac{3}{2}}}P + \frac{1}{8\pi^2}H^{\frac{3}{2}}\partial_j H^{\frac{3}{2}}P + (3H)^{-1}\partial_j V\right)P. \tag{13}$$

We express $H(\phi)$ as a function of $\phi$ from Equation (5). The dependence of $a$ on $\phi$ is more involved. We determine it in the slow-roll approximation in Equation (10). All the formulas in this section can be generalized to a multifield case (Equation (11)) just by a replacement $\phi \to |\phi|$. For the double-well potential (treated as an approximate realization of the models of inflation [36,37], e.g., as an approximation to the Starobinsky model [38,39]),

$$V(\phi) = \frac{g}{4}(\phi^2 - \frac{\mu^2}{g})^2, \tag{14}$$

$$a = |\phi|^{\frac{2\pi G\mu^2}{g}} \exp(-\pi G\phi^2). \tag{15}$$

If $a \to 0$, then either $\phi \to 0$ or $\phi \to \infty$. According to Equation (9) ($B = W = 0$), the classical slow roll time evolution of $\phi$ is determined by

$$- \sqrt{24\pi G} \left( \frac{g}{4} (\phi^2 - \frac{\mu^2}{g})^2 \right)^{\frac{1}{2}} \frac{d\phi}{dt} = g\phi(\phi^2 - \frac{\mu^2}{g}). \tag{16}$$

Hence, if $\mu g^{-\frac{1}{2}} \geq \phi \geq 0$, then $\phi$ is increasing. If $\phi \geq \mu g^{-\frac{1}{2}}$, then $\phi$ is decreasing to $\mu g^{-\frac{1}{2}}$. It follows that $\phi(t) \to \mu g^{-\frac{1}{2}}$. In reality, the ground state is beyond the slow roll regime as the slow-roll parameters

$$\tilde{\epsilon} = \frac{1}{16\pi G} \left( \frac{V'}{V} \right)^2 = \frac{1}{16\pi G} \phi^2 (\phi^2 - \frac{\mu^2}{g})^{-2}$$

and

$$\tilde{\eta} = \frac{1}{8\pi G} \frac{V''}{V} = \frac{1}{2\pi G} (3\phi^2 - \frac{\mu^2}{g})(\phi^2 - \frac{\mu^2}{g})^{-2}.$$

tend to infinity close to the ground state. Nevertheless, the stochastic dynamical system in Equation (9) still makes sense close to the minimum of the potential but presumably does not approximate the second-order wave Equation (6). As shown in Section 5 for double-well potentials (for a general discussion, see [22]), the stochastic equation with the Starobinsky noise leads to a non-integrable stationary probability distribution (if treated as a system on the whole real line), whereas the system with the thermal noise has a normalizable stationary distribution. The fact that the solution of the equation for the stationary probability distribution is not normalizable could be ignored. It appears in a range of $\phi$ where the restrictions on the parameters $\tilde{\epsilon}$ and $\tilde{\eta}$ are violated. For deterministic systems, we can restrict the initial values of the field and the time evolution in order to satisfy the requirement of small $\tilde{\epsilon}$ and $\tilde{\eta}$. It is more difficult to do it in a random system because the noise can move the system to the forbidden region of the field configurations. We can define the stochastic system in a required domain of field configurations by an imposition of boundary conditions (as discussed in [34,40] for the Starobinsky stochastic equation with $\gamma = 0$). However, in such a case, the stochastic process and its probability distribution depend on the boundary conditions. Nevertheless, some correlation functions may have a negligible dependence on boundary conditions, as discussed in [34,40] (for more on boundaries in diffusions, see [31]). As a result of fluctuations, the stochastic process does not feel the singularity $g\phi^2 = \mu^2$. The requirement of small values of $\tilde{\epsilon}$ and $\tilde{\eta}$ can be treated as the requirements of a strong friction which is violated when $H \simeq |\phi^2 - \frac{\mu^2}{g}|$ in Equation (6). The quantum noise in Equation (7) (which is multiplied by $H$) is also vanishing at $H(\phi) = 0$. It is known in the theory of Brownian motion [41] that under the assumption of a strong friction the random second-order differential Equation (6) can be replaced by the first-order Equation (9). It may remain true with the thermal noise in Equation (7) which is not vanishing. In any case, the presence of the thermal noise allows an asymptotic regular behavior of the solutions of $\phi_t$, as is exhibited by the existence of the stationary probability (proved in Section 6). Hence, we expect that the stochastic slow-roll process in Equation (9) may well approximate the second-order one despite the increase of the small-roll parameters $\tilde{\epsilon}$ and $\tilde{\eta}$ at $g\phi^2 = \mu^2$.

Equation (16) is non-analytic at $g\phi^2 = \mu^2$. As an example of a potential which does not lead to a singular slow-roll equation, we consider

$$V(\phi) = \frac{g_4}{4} (\phi^2 - \frac{\mu^2}{g})^4, \tag{17}$$

where $g_4$ is a dimensional constant. Then,

$$a(\phi) = |\phi|^{\frac{\pi G \mu^2}{g}} \exp(-\frac{1}{2} \pi G \phi^2). \tag{18}$$

## 4. Expectation Value of the Energy-Momentum in the Semi-Classical Approximation

In Equation (1), it is understood that the evolution equation for the quantum field $\phi_t$ has been solved. Such an equation can be formulated by imposing the canonical commutation relations at the initial time. In static coordinates, it may be possible to define a Hamiltonian which generates the time evolution (and allows defining a thermal state). If there is a Hamiltonian $\mathcal{H}$ generating a unitary time evolution $\exp(-i\mathcal{H}t)$(at least at small time), then, decomposing the expectation value in Equation (1) in a complete set of eigenfunctions $|E>$ of $\mathcal{H}$, we get

$$
\begin{aligned}
&< \psi|V(\phi_t(\phi))|\psi > = < \psi_t|V(\phi)|\psi_t >\\
&= \int_0^\infty dE \int_0^\infty dE' < \psi|E >< E|V(\phi)|E' >< E'|\psi > \exp(-i(E'-E)t)\\
&= \int_{-\infty}^\infty d\epsilon\, \gamma(\epsilon) \exp(-i\epsilon t),
\end{aligned}
\tag{19}
$$

where we introduce

$$
\epsilon = E' - E
$$

and

$$
\gamma(\epsilon) = \int_0^\infty dE < E + \epsilon|\psi >< \psi|E >< E|V(\phi)|E + \epsilon > .
\tag{20}
$$

Here, $|E>$ is the eigenstate of the Hamiltonian with an eigenvalue $E \geq 0$. The behavior of the Fourier transform depends on $\gamma(\epsilon)$. Note, however, that $\epsilon$ is unbounded from below and the Paley–Wiener theorem is not applicable. We could also write Equation (19) in the form

$$
< \psi|V\Big(\phi_t(\phi)\Big)|\psi > = \sum_{n,m} < \psi_t|\chi_n >< \chi_m|\psi_t >< \chi_n|V(\phi)|\chi_m >,
\tag{21}
$$

where $\chi_n$ is a basis of eigenstates. Then, the Paley–Wiener theorem can be applied estimating each term (as in [42,43]) in the series in Equation (21) but the behavior of the infinite sum remains unclear. The integrals in Equation (20) are calculable in quantum mechanics if we solve the eigenvalue problem in a given potential $V$. However, it is rather difficult to determine the time behavior of the Fourier integrals in general. Equation (19) may be problematic in the case of a time dependent metric. Nevertheless, we can see that the exponential decay in Equation (19) (which we prove below) is not forbidden by the Paley–Wiener theorem even in an ideal case of the Minkowski metric.

We calculate the expectation value of the energy-momentum in some approximations. We hope that it correctly describes the relevant regime of the vacuum decay. We consider the potential in Equation (14). Then, Equation (16) can be expressed as

$$
- \sqrt{6\pi G g}\frac{d\phi}{dt} = g\phi\epsilon(g\phi^2 - \mu^2),
\tag{22}
$$

where $\epsilon$ is an antisymmetric function with $\epsilon(y) = 1$ for $y > 0$ and $\epsilon(0) = 0$. Equation (22) is discontinuous at the minimum of the potential. Let

$$
Y(\phi) = \phi^2 - \frac{\mu^2}{g}
\tag{23}
$$

and

$$
\alpha = \sqrt{\frac{g}{6\pi G}}.
$$

Then, Equation (22) reads

$$
\partial_t Y = -2\alpha Y\epsilon(Y) - \frac{2\alpha\mu^2}{g}\epsilon(Y)
$$

with the solution $\phi_t(\phi)$ for $g\phi^2 > \mu^2$

$$\phi_t(\phi)^2 = \exp(-2\alpha t)\phi^2 \tag{24}$$

as long as $\exp(-2\alpha t)\phi^2 \geq \frac{\mu^2}{g}$ and the solution $\phi_t(\phi)$ for $g\phi^2 < \mu^2$

$$\phi_t(\phi)^2 = \exp(2\alpha t)\phi^2 \tag{25}$$

as long as $\exp(2\alpha t)\phi^2 \leq \frac{\mu^2}{g}$. Let us note that a solution exists only for a limited time interval $[0, t]$ (depending on $\phi$)

$$\exp(2\alpha t) \leq \frac{\mu^2}{g\phi^2}$$

if $g\phi^2 < \mu^2$ and

$$\exp(-2\alpha t) \geq \frac{\mu^2}{g\phi^2}$$

if $g\phi^2 > \mu^2$.

The solutions in Equations (24) and (25) exist until the time when $\phi_t$ achieves the minimum. However, when $g\phi^2$ is close to $\mu^2$, the slow roll conditions of small $\tilde{\epsilon}$ and small $\tilde{\eta}$ are violated. Hence, close to the minimum of the potential, the solution $\phi_t(\phi)$ does not approximate the solutions of the wave in Equation (6). In fact, the solutions $\phi_t(\phi)$ of Equation (6) oscillate in the vicinity of the minimum of the potential, whereas Equations (24) and (25) decay to the minimum. We can check whether the second time derivative in Equation (6) is negligible, i.e., if $|\partial_t^2\phi| << 3H|\partial_t\phi|$. From Equations (24) and (25), we obtain that this is the case if $\alpha << 3H$, i.e., the decay of $\phi_t$ should be slow in comparison to the Hubble expansion. From Equation (5), we obtain

$$|\phi^2 - \frac{\mu^2}{g}| >> \frac{1}{6\pi G} = \frac{4}{3}m_{PL}^2. \tag{26}$$

Comparing with the definitions of $\tilde{\epsilon}$ and $\tilde{\eta}$, we can see that $\tilde{\epsilon}$ and $\tilde{\eta}$ will be small if $\phi^2 << m_{PL}^2$ and if the condition in Equation (26) is satisfied or if $\frac{\mu^2}{g} >> m_{PL}^2$.

We calculate a decay of the expectation value of the energy-momentum in a state $\psi$ which is not a ground state. Let us consider a wave function concentrated at $\phi = 0$

$$\psi = \left(\frac{\sigma}{2\pi}\right)^{\frac{1}{4}} \exp(-\frac{\sigma\phi^2}{4}) \tag{27}$$

with a certain $\sigma > 0$. The expectation value in Equation (19) is

$$< \psi | V\left(\phi_t(\phi)\right) | \psi > = \left(\frac{2\pi}{\sigma}\right)^{\frac{1}{2}} \frac{g}{4} \left( \int_{\sqrt{\frac{1}{g}}\exp(\alpha t)\mu}^{\infty} d\phi \exp(-\frac{\sigma\phi^2}{2})(\exp(-2\alpha t)\phi^2 - \frac{\mu^2}{g})^2 \right.$$
$$\left. + \int_0^{\sqrt{\frac{1}{g}}\exp(-\alpha t)\mu} d\phi \exp(-\frac{\sigma\phi^2}{2})(\exp(2\alpha t)\phi^2 - \frac{\mu^2}{g})^2 \right), \tag{28}$$

where as $\phi_t(\phi)$ we insert the solutions in Equations (24) and (25). We can express the first integral in Equation (28) by the probability function $\Phi$ [44]

$$\int_v^{\infty} du \exp(-\frac{u^2}{4\beta}) = \sqrt{\pi\beta}\left(1 - \Phi(\frac{v}{2\sqrt{\beta}})\right). \tag{29}$$

It decays more quickly than exponentially. The second integral can be estimated by an elementary change of variables. We obtain

$$< \psi | V(\phi_t(\phi)) | \psi > = < \psi | V(\phi) | \psi > \exp(-\alpha t)(1 + K(t)) \tag{30}$$

(with a bounded function $K(t), K(0) = 0$) showing the exponential decay of the cosmological term.

Next, let us consider a wave function $\chi$ concentrated around the ground state at $|\phi| = \frac{\mu}{\sqrt{g}}$ such that

$$\chi = const \tag{31}$$

in the interval

$$\frac{\mu}{\sqrt{g}} - r \leq |\phi| \leq \frac{\mu}{\sqrt{g}} + r \tag{32}$$

with a certain $r \simeq m_{PL}$ (according to Equation (26)) and $\chi = 0$ outside this interval. Then, for sufficiently small r (depending on $t$), the only solution of the classical equation is the constant solution with the initial condition $|\phi| = \frac{\mu}{\sqrt{g}}$. The contribution of this solution to the expectation value is

$$< \chi_t | V(\phi) | \chi_t > = 0. \tag{33}$$

We can conclude that the expectation value of the energy in a state concentrated at the false vacuum of $\phi$ is decaying in time, whereas the expectation value concentrated at the ground state value of $\phi^2$ (at $\frac{\mu^2}{g}$) is zero.

It is not straightforward to compare the result in Equation (30) with other studies of the quantum scalar field in expanding universes. This has been usually done in de Sitter space with $H = const$, whereas, in our Einstein–Klein–Gordon system, $H$ is determined by the field $\phi_t$ in Equation (5). Note that if (according to Equation (24)) $\phi_t^2 = \exp(-2\alpha t)\phi^2$ for $g\phi^2 > \mu^2$, then $H$ is decreasing exponentially to zero. The same is true for $g\phi^2 < \mu^2$ when $\phi_t^2 = \exp(2\alpha t)\phi^2$ (according to Equation (25)). We can compare our results with the ones concerning quantum field theory with $H = const$. Thus, the decay rate $\alpha$ of Equations (24) and (30) coincides with the one of Starobinsky and Yokoyama ([30], Equation (60)) if in Equation (5) for $H$ we insert $\phi = 0$ (the local maximum) of the potential.

Equation (16) being piece-wise linear may look rather special. For the potential in Equation (17), we obtain the slow-roll equation

$$\frac{d\phi}{dt} = -2\alpha_4 \phi(\phi^2 - \frac{\mu^2}{g}), \tag{34}$$

where

$$\alpha_4 = \sqrt{\frac{g_4}{6\pi G}}.$$

Its solution with the initial condition $\phi$ is

$$\phi_t(\phi)^2 = \frac{\mu^2}{g} \phi^2 \left( \phi^2 (1 - \exp(-\frac{4\alpha_4 \mu^2 t}{g})) + \frac{\mu^2}{g} \exp(-\frac{4\alpha_4 \mu^2 t}{g}) \right)^{-1}. \tag{35}$$

When we insert Equation (35) into the expectation value in Equation (19) with the Gaussian wave function in Equation (27) concentrated at $\phi = 0$, there will be an unphysical contribution from an unstable fixed point $\phi_t$ with the initial condition $\phi \simeq 0$. To eliminate this contribution, we assume that $\psi(\phi) = 0$ for $|\phi| \leq r$; then, we obtain an estimate

$$< \psi | V(\phi_t) | \psi > \leq K(r) \exp(-\frac{16\alpha_4 \mu^2 t}{g}). \tag{36}$$

If $\psi$ is sharply concentrated at $g\phi^2 = \mu^2$ with the variance $\frac{1}{\sigma} \to 0$, then

$< \psi_t | V(\phi) | \psi_t > \simeq \frac{1}{\sigma^4} \to 0$. In the next sections, we calculate the expectation values resulting from quantum and thermal fluctuations at large time.

## 5. Expectation Value of the Energy-Momentum with the Stochastic Slow Roll Approximation

We neglect the second-order time derivatives but we take into account the quantum and thermal noise. Now, Equation (9) for "quantum" $\phi$ in a thermal state reads

$$\sqrt{6\pi g G}\frac{d\phi}{dt} = -g\phi\epsilon(g\phi^2 - \mu^2) + \gamma a^{-\frac{3}{2}}|\phi^2 - \frac{\mu^2}{g}|^{-1}\frac{dB}{dt} + \frac{3}{2\pi}|\phi^2 - \frac{\mu^2}{g}|^{\frac{3}{2}}(\frac{8\pi G g}{3})^{\frac{5}{4}}\frac{dW}{dt}. \quad (37)$$

The approximations of this equation depend on whether $\phi$ is close to the minimum of $V$ or not. For a small $\phi$ ($\phi^2 << \frac{\mu^2}{g}$), we have

$$d\phi = \alpha\phi dt + \frac{\alpha\gamma}{\mu^2}a^{-\frac{3}{2}} \circ dB + \frac{3\alpha\mu^3}{2\pi}(\frac{8\pi G}{3g})^{\frac{5}{4}} \circ dW. \quad (38)$$

From Equation (15), $a^{-\frac{3}{2}} \simeq \phi^{-\frac{3\pi G\mu^2}{g}}$. Hence, this term is large for a small $\phi$, whereas the last term in Equation (38) is negligible in comparison to the thermal one. Let us consider $Q = \phi^m$. Then,

$$dQ = m\phi^{m-1} \circ d\phi = \alpha m Q dt + \frac{\alpha\gamma}{\mu^2}mQ^{\frac{m-1-\frac{3\pi G\mu^2}{g}}{m}} \circ dB. \quad (39)$$

If we choose

$$m = 1 + \frac{3\pi G\mu^2}{g}, \quad (40)$$

then $Q$ is the Ornstein–Uhlenbeck process [31]

$$Q_t = \exp(m\alpha t)\phi^m + \frac{\alpha\gamma}{\mu^2}m\int_0^t \exp(m\alpha(t-s))dB_s. \quad (41)$$

Equation (38) allows calculating $|\phi| = |Q|^{\frac{1}{m}}$. From the correlation functions of the Ornstein–Uhlenbeck process, we can conclude that the classical behavior $\phi \simeq \exp(\alpha t)$ is strengthened by the noise. This can already be seen when calculating the covariance of $Q_t$

$$\langle Q_t^2 \rangle = \exp(2m\alpha t)\phi^{2m} + \frac{m\gamma^2}{2\mu^4\alpha}(\exp(2m\alpha t) - 1). \quad (42)$$

We can also calculate the expectation value in a thermal state $\rho$ using the formula

$$Tr\Big(\rho V\big(\phi_t(\phi)\big)\Big) = \int d\phi d\phi' P_0(\phi)P_t(\phi,\phi')V(\phi'), \quad (43)$$

where $P_t$ is the transition function for the stochastic process $\phi_t$ (Equation (38)) (see [33] for Equation (43)). $P_0$ is the distribution of the initial value. For a thermal state $\rho$ and the Ornstein–Uhlenbeck $Q$, we would have

$$P_0(Q) = \exp(-\frac{m\alpha}{\hbar}Q\tanh(\frac{\hbar\beta m}{2}\alpha)Q),$$

where $\beta$ is the inverse temperature. We would have to express $\phi_t$ by $Q_t$ in Equation (43) in order to obtain the transition function for $\phi_t$ in terms of the known transition function for the Ornstein–Uhlenbeck process $Q_t$. With the result in Equation (42), the calculations of Equation (43) confirm the growth of $|\phi|$ from its small initial value.

If $g\phi^2 >> \mu^2$, then there is a damping negative power of $\phi$ in $a^{-\frac{3}{2}}$ in the thermal term (which is dominating at large $\frac{\mu^2}{g}$ before the exponential in Equation (15) becomes significant). When we neglect the thermal term in Equation (37), the stochastic equation reads

$$d\phi = -\alpha\phi dt + \lambda\phi^3 \circ dW,$$ (44)

where

$$\lambda = \frac{3\alpha}{2\pi g}\left(\frac{8\pi Gg}{3}\right)^{\frac{5}{4}}.$$

Let

$$\Omega = \phi^{-2}.$$

Then,

$$d\Omega = 2\alpha\Omega dt - 2\lambda dW.$$ (45)

Hence,

$$\Omega_t = \phi^{-2}\exp(2\alpha t) - 2\lambda\int_0^t \exp(2\alpha(t-s))dW_s.$$ (46)

The expectation value of the potential could again be determined by the rhs of Equation (43) by expressing $\phi_t$ by $\Omega_t$ and calculating the expectation values of $\Omega_t$ by means of the known transition function for the Ornstein–Uhlenbeck process (as a result, $\phi_t \simeq \exp(-\alpha t)$). In the limit $\lambda \to 0$, the transition function $P_t$ tends to the $\delta$ function. Then, with $P_0 = |\psi|^2$, we return to Equation (30) for the expectation value of the previous section. In another way, we can see that the quantum noise supports the classical behavior derived in Section 5 as from Equation (46) it follows (we can derive such expectation values for any power of $\Omega_t$)

$$\langle\Omega_t^2\rangle = \phi^{-4}\exp(4\alpha t) + \frac{\lambda^2}{\alpha}(\exp(4\alpha t) - 1).$$ (47)

According to Equation (47), $\phi_t^{-2}$ is growing in time; hence, $\langle V \rangle$ is decreasing in time. As shown in the next section, it is decreasing to the expectation value in a stationary state. Equations (42) and (47) show that the corrections resulting from thermal as well as quantum fluctuations lead to the same decay law $\exp(-2\alpha t)$ (appearing also in [30], Section IVC). Only the amplitude of $\phi$ fluctuation is adding to the classical $\phi^2$.

The stochastic equation for the potential in Equation (17) takes the form

$$d\phi = -2\alpha_4\phi\left(\phi^2 - \frac{\mu^2}{g}\right) + \frac{\alpha_4\gamma}{g}a^{-\frac{3}{2}}\left(\phi^2 - \frac{\mu^2}{g}\right)^{-2} \circ dB$$
$$+ \frac{3\alpha_4}{2\pi g}\left(\frac{2\pi gG}{3}\right)^{\frac{5}{4}}|\phi^2 - \frac{\mu^2}{g}|^3 \circ dW.$$ (48)

For small $\phi$, the thermal noise proportional to $a^{-\frac{3}{2}}$ dominates over the quantum noise. We can write an approximation of Equation (48) using Equation (18) for $a^{-\frac{3}{2}}$ (neglecting the exponential for a small $\phi$) as

$$d\phi = \frac{2\alpha_4\mu^2}{g}\phi dt + \frac{\alpha_4\gamma g}{\mu^4}|\phi|^{-\frac{3\pi G\mu^2}{2g}} \circ dB.$$ (49)

Let

$$q = |\phi|^r$$ (50)

with

$$r = 1 + \frac{3\pi G\mu^2}{2g}$$ (51)

Then,

$$dq = \frac{2\alpha_4 r\mu^2}{g} q\,dt + \frac{r\alpha_4 \gamma g}{\mu^4} dB. \tag{52}$$

The solution of Equation (52) is the Ornstein–Uhlenbeck process. We can calculate all correlation functions as in Equation (42) and conclude that, for a small $\phi$, we have $\phi_t(\phi)^2 \simeq \exp(\frac{4\alpha_4\mu^2}{g}t)$. This conclusion is in agreement with the classical solution in Equation (35) when we expand it in $\phi^2$.

## 6. Fokker–Planck Equation and Its Stationary Probability Distribution

Environmental noise is present in all physical systems. Its crucial role in equilibration of dynamical systems is well-known [45]. Its action can be seen as a stabilization. Some correlation functions have a singular or chaotic behavior at $t \to \infty$ without noise and a smooth behavior with noise. We could approach the calculation of the expectation value of the energy-momentum by means of the Fokker–Planck equation for the transition probability (12). When the initial state is far from the ground state, we have the exponential decay of expectation values in the semi-classical approximation in Section 4. As a result of thermal and quantum fluctuations, there will be a non-zero correction to this result. It is rather difficult to obtain an exact solution of the Fokker–Planck equation (Equation (12)) or even to estimate a qualitative behavior of its solutions. However, the large time limit can be obtained from the equation for the stationary distribution.

The stationary probability $P_\infty(\phi)$ is the limit of $P_t$ for $t \to \infty$ [35] (it does not depend on the initial condition). It can be obtained from the requirement $\partial_t P = 0$, which gives (with the Stratonovitch interpretation and after an integration over $\phi$)

$$\frac{\gamma^2}{18} \frac{1}{Ha^{\frac{3}{2}}} \partial_\phi \frac{1}{Ha^{\frac{3}{2}}} P + \frac{1}{8\pi^2} H^{\frac{3}{2}} \partial_\phi H^{\frac{3}{2}} P + (3H)^{-1} V' P = 0. \tag{53}$$

In the multidimensional case in Equation (13), the requirement $\partial_t P = 0$ is satisfied (for functions $P$ and $V$ depending only on $|\phi|$) if

$$\frac{\gamma^2}{18} \frac{1}{Ha^{\frac{3}{2}}} \partial_{|\phi|} \frac{1}{Ha^{\frac{3}{2}}} P + \frac{1}{8\pi^2} H^{\frac{3}{2}} \partial_{|\phi|} H^{\frac{3}{2}} P + (3H)^{-1} P \partial_{|\phi|} V = 0. \tag{54}$$

Let us consider the simplest case first. The stationary solution of Equation (53) without the Starobinsky (quantum) noise is

$$\begin{aligned} P_\infty &\equiv \sqrt{V} a^{\frac{3}{2}} \exp(-\gamma^{-2} F(\phi)) = \sqrt{V} \exp\left(-12\pi G \int^\phi d\phi'(V')^{-1} V\right) \\ &\times \exp\left(-\frac{6}{\gamma^2} \sqrt{\frac{8\pi G}{3}} \int d\phi V' \sqrt{V} \exp(-24\pi G \int^\phi d\phi'(V')^{-1} V)\right). \end{aligned} \tag{55}$$

For the potential in Equation (14)

$$P_\infty = |\phi^2 - \frac{\mu^2}{g}| |\phi|^{\frac{3\pi G\mu^2}{g}} \exp(-\frac{3}{2}\pi G\phi^2) \exp(\frac{-F(\phi)}{\gamma^2}), \tag{56}$$

where

$$F(\phi) = \int^\phi U(\phi) \tag{57}$$

and

$$U(\phi) = 6\sqrt{\frac{8\pi G}{3}} V' \sqrt{V} \exp\left(-24\pi G \int^\phi d\phi'(V')^{-1} V\right).$$

We can calculate now the expectation value of $V$ (the cosmological term $\Lambda$)

$$\Lambda = \langle V \rangle = \left(\int P_\infty\right)^{-1} \int P_\infty V(\phi) \tag{58}$$

for a small $\gamma$ by means of the saddle-point method. The critical points are determined by $F'(\phi_c) = U(\phi_c) = 0$), i.e.,

$$V'\sqrt{V}\exp\left(-24\pi G\int^{\phi}d\phi'(V')^{-1}V\right) = 0. \tag{59}$$

For the potential in Equation (14), $\phi_c = 0$ is the maximum of $F$. The minima of $F$ are at $\phi_c = \pm\frac{\mu}{\sqrt{g}}$. We shift the variables in the integrals in the nominator and the denominator in Equation (58)

$$\phi = \phi_c + \gamma^{\nu}q \tag{60}$$

with a certain $\nu.\nu \neq 1$ is chosen in such a way that $\gamma^2$ in the denominator of the exponent in Equation (56) cancels with $\gamma$-dependent terms in the expansion of $F$ in $q$. For the potential in Equation (14) $\nu = \frac{2}{3}$. Expanding Equation (58) in $\gamma$, we obtain

$$\Lambda = \gamma^{\frac{4}{3}}K \tag{61}$$

with a certain constant $K(\gamma) > 0$ for any $\gamma$. The functions $F(\phi)$ as well as $U(\phi)$ are non-analytic at $g\phi^2 = \mu^2$. If we consider the potential in Equation (17) or more general $(\phi^2 - \frac{\mu^2}{g})^{4n}$ with a natural number $n$, then Equation (61) still holds true. It is remarkable that the index $\frac{4}{3}$ appears also in Kolmogorov's theory of turbulence as a relation between the dissipation scale and viscosity. For general potentials $V$, if $U'(\phi_c) \neq 0$, then $\Lambda \simeq \gamma^2$. We obtain the result that $\Lambda$ is proportional to the diffusion constant $\gamma^2$ in a different model with a diffusive fluid in the energy-momentum tensor of Einstein equations [46].

If $\gamma = 0$ (the environmental noise is absent), then we obtain the Starobinsky solution [18](discussed also by Linde [47]), which in the Stratonovitch interpretation of the stochastic Equation (37) takes the form

$$P_{\infty} = V^{-\frac{3}{4}}\exp(\frac{3}{8G^2}\frac{1}{V}). \tag{62}$$

However, because $V \to 0$ when $\phi^2 \to \frac{\mu^2}{g}$, this is not a stationary distribution because it is not integrable at $\phi^2 = \frac{\mu^2}{g}$. The point $g\phi^2 = \mu^2$ is beyond the slow-roll approximation. To restrict Equation (37) at $\gamma = 0$ to the inflationary domain, we would need to cut the range of $\phi$ by imposing the boundary conditions as in [34]. We show here that with the thermal noise we can get stationary solutions without imposing boundary conditions.

With $\gamma \neq 0$ and the quantum noise, we write

$$\tilde{P} = H^{-1}a^{-\frac{3}{2}}P. \tag{63}$$

Then, Equation (53) for $\tilde{P}$ is

$$\frac{\gamma^2}{18}H^{-1}a^{-\frac{3}{2}}\partial_{\phi}\tilde{P} + \frac{1}{8\pi^2}H^{\frac{3}{2}}\partial_{\phi}(H^{\frac{5}{2}}a^{\frac{3}{2}}\tilde{P}) = -\frac{1}{3}V'a^{\frac{3}{2}}\tilde{P}. \tag{64}$$

Using the formulas for $H$ and $a$ (Equation (10)), we obtain

$$\begin{aligned}\ln\tilde{P} = &-6\int d\phi Ha^3(\gamma^2 + \frac{9}{4\pi^2}H^5a^3)^{-1}\\&\times\left(V' + \frac{10}{3}G^2VV' - 32\pi G^3V^3(V')^{-1}\right).\end{aligned} \tag{65}$$

From Equation (15), we have $a \to 0$ exponentially for $\phi \to \infty$, then $a^3H^5 \to 0$, and we get from Equations (63)–(65) a formula coinciding for large $\phi$ with Equation (55) (the thermal noise is dominating). We obtain a stationary solution which is integrable at large $\phi$. It can be seen that there is

no integrability problem at $\phi = 0$ as well as at $\phi^2 = \frac{\mu^2}{g}$. For a small $\gamma$ and $a^3 H^5 \to 0$, Equation (65) is of the same form as Equation (55).

As the stationary distribution gives a non-zero weight to the fluctuations around $\phi^2 = \frac{\mu^2}{g}$, we obtain a non-zero cosmological constant at arbitrary time. When $V(\phi)$ is large at $\phi \simeq 0$, the cosmological term will be large at early time when the stochastic process starts near $\phi \simeq 0$, small when $\phi_t(\phi)^2$ is approaching $\phi^2 = \frac{\mu^2}{g}$, and at large time it is approaching the limit in Equation (61), which determines a small $\Lambda$ for a small thermal dissipation $\gamma$. The diffusion constant $\gamma^2$ is proportional to the temperature at the end of the radiation era in warm inflation models (this can be the intrinsic temperature of the de Sitter space).

## 7. Summary and Conclusions

The cosmological term can be interpreted as the expectation value of the energy density in a quantum state describing the universe evolution. In the early stage of inflation, it is large when the quantum state of the universe is far from the ground state. At later stage, the time evolution moves the state closer to the ground state, leading to the small value of the energy density. The final value of the cosmological term is different from zero because of the thermal and quantum fluctuations at the end of inflation.We derive such a behavior of the cosmological term on the basis of some reasonable assumptions in the model of inflation governed by a scalar quantum field (inflaton) described by the double-well potential. The form of the potential is not essential for the conclusions. We could repeat the calculations for any polynomial double-well potential. The asymptotic value of the cosmological term is in a unique way determined by the stationary distribution. The fluctuations which determine the asymptotic value of the cosmological term depend on the asymptotic probability distribution of the stochastic process governing the inflaton. When inflation ends, the thermal noise overcomes the quantum noise. As a result, the fluctuations (defining the cosmological constant) are proportional to the thermal diffusion constant, as happens in most models of the diffusion processes. If at the final de Sitter stage an equilibrium is achieved, then by the dissipation–fluctuation theorem the diffusion constant is proportional to the final temperature. If we had a good candidate for the initial state of the universe, we could obtain the precise time evolution of the cosmological term until its final stationary value.

We study the inflaton contribution to the energy-momentum as the vacuum energy $< V >$ in the long wave limit (consistent with the slow-roll approximation). The expectation values of the derivative terms in the energy-momentum at high momenta of quantum field theory need renormalzation. The renormalized energy-momentum contributes (to the lhs of Einstein equations) terms quadratic in the curvature [28,29] (it can also give a divergent cosmological term leading to the cosmological constant problem discussed in [14]). Such terms modify equations of motion (can lead to exponential expansion of $a(t)$ [29]). In some models of this type, the quadratic terms can be expressed by an extra scalar field. We obtain an effective field theory of Einstein gravity [38,39] similar to the one studied in this paper. We think that the model can accommodate some quantum effects (long wave fluctuations in the $\delta N$-formalism [25,26] and short wave fluctuations by transformation [38,39]) as well as the classical $\Lambda CDM$ evolution.

**Funding:** This research received no external funding

**Conflicts of Interest:** The authors declare no conflict of interest.

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
