# Peer review of "Semi-Classical Einstein Equations: Descend to the Ground State"

_universe, doi:10.3390/universe6060074_

Round 1
Reviewer 1 Report
This article deals with a quantum approach to the vacuum energy driving inflation and the present day cosmological constant. While the approach followed may be interesting,
I believe the author should clarify important points.
An appealing aspect of the authors approach is a unified treatment of the background energy during inflation and nowadays. While this paper contains interesting, it raises several questions.
a) In the standard inflationary calculations, the background space-time is treated classically with energy densities well below the Planckian values. Only the fluctuations are treated quantummechanically as they arise from quantum fluctuations. In the author's framework it is unclear if and how the two are separated, and what are thepredictions in this setup.
b) I suggest the author add the following important references concerning the calculation
of inflationary perturbations:
S.W. Hawking, Phys.Lett.B 115, 295 (1982)
A.A. Starobinsky, Phys.Lett.B 117, 175 (1982)
c) I suggest to add the following references about their classicality today:
A. Albrecht, P. Ferreira, M. Joyce, T. Prokopec, Phys.Rev.D 50, 4807 (1994)
D. Polarski, A.A. Starobinsky, Class.Quant.Grav. 13, 377 (1996)
d) It is not clear how the inflationary perturbations are taken into account if their mean value vanishes
Reviewer 2 Report
In the manuscript, the authors consider Einstein' equations with
the expectation values of the energy-momentum tensor in some
quantum state on the right-hand side. The proposed aims are to
investigate the time evolution of the energy density in a state
simulating the false vacuum, compare it with the energy density
in a state close to the true ground state, and to analyze the
behavior of the cosmological term during a transition from the
false vacuum to the ground state. For this purpose, the authors
calculate an expectation value of the stress-energy tensor and
show that in the false vacuum it exponentially decreases with time.
This is made using the classical and slow-roll approximations which,
however, are not reliable at large times. The large-time behavior
is determined by solving the equation for a stationary probability
distribution. This allows to calculate the expectation values in
the large time limit. The obtained results could help to understand
why the magnitude of the cosmological term decreases from some big
value at the epoch of inflation to a relatively small value at the
present epoch in the evolution of the Universe.
I have several comments to be taken into account:
1. An equation for the metric with no number [in between Eqs.
(5) and (6)] should be explained in more detail. Is it SPATIALLY
flat metric? Whether the scale factor $a$ is a constant or it can
depend on $t$?
2. Equation (55) is unclear. What sign (+, -, x) should stay in
front of the second line?
3. Equation (65) is unclear for the same reason.
4. All the results were obtained for the model of inflation
governed by a scalar quantum field. In the conclusion section
some remark is welcome whether similar results are obtainable
for the models of inflation governed by the vacuum polarization
of standard quantum fields [see S.G.Mamaev and V.M.Mostepanenko,
Zh. Eksp. Teor. Fiz. v.78, 20 (1980) (Sov. Phys. JETP v.51, 9 (1980));
A.A.Starobinsky, Phys. Lett. B v.91, 99 (1980)].
5. The punctuation signs after many equations are missing, see, e.g.,
equation after Eq. (10), Eqs. (19), (20), (21), (28), etc., and
should be added.
Round 2
Reviewer 1 Report
I have seen the revised version. The author has amended it in a way that I can recommend it for publication.
Reviewer 2 Report
In the revised manuscript, all comments contained in my first report have been satisfactorily taken into account. The revised manuscript is recommended for publication.